# Parallel Predictive Entropy Search for Batch Global Optimization of Expensive Objective Functions

**Amar Shah**
Department of Engineering
Cambridge University
as793@cam.ac.uk

**Zoubin Ghahramani**
Department of Engineering
University of Cambridge
zoubin@eng.cam.ac.uk

## Abstract

We develop *parallel predictive entropy search* (PPES), a novel algorithm for Bayesian optimization of expensive black-box objective functions. At each iteration, PPES aims to select a *batch* of points which will maximize the information gain about the global maximizer of the objective. Well known strategies exist for suggesting a single evaluation point based on previous observations, while far fewer are known for selecting batches of points to evaluate in parallel. The few batch selection schemes that have been studied all resort to greedy methods to compute an optimal batch. To the best of our knowledge, PPES is the first non-greedy batch Bayesian optimization strategy. We demonstrate the benefit of this approach in optimization performance on both synthetic and real world applications, including problems in machine learning, rocket science and robotics.

## 1 Introduction

Finding the global maximizer of a non-concave objective function based on sequential, noisy observations is a fundamental problem in various real world domains e.g. engineering design [1], finance [2] and algorithm optimization [3]. We are interesed in objective functions which are unknown but may be evaluated pointwise at some *expense*, be it computational, economical or other. The challenge is to find the maximizer of the expensive objective function in as few sequential queries as possible, in order to minimize the total expense.

A Bayesian approach to this problem would probabilistically model the unknown objective function, $f$. Based on posterior belief about $f$ given evaluations of the the objective function, you can decide where to evaluate $f$ next in order to maximize a chosen utility function. *Bayesian optimization* [4] has been successfully applied in a range of difficult, expensive global optimization tasks including optimizing a robot controller to maximize gait speed [5] and discovering a chemical derivative of a particular molecule which best treats a particular disease [6].

Two key choices need to be made when implementing a Bayesian optimization algorithm: (i) a model choice for $f$ and (ii) a strategy for deciding where to evaluate $f$ next. A common approach for modeling $f$ is to use a Gaussian process prior [7], as it is highly flexible and amenable to analytic calculations. However, other models have shown to be useful in some Bayesian optimization tasks e.g. Student-$t$ process priors [8] and deep neural networks [9]. Most research in the Bayesian optimization literature considers the problem of deciding how to choose a single location where $f$ should be evaluated next. However, it is often possible to probe several points *in parallel*. For example, you may possess 2 identical robots on which you can test different gait parameters in parallel. Or your computer may have multiple cores on which you can run algorithms in parallel with different hyperparameter settings.

Whilst there are many established strategies to select a single point to probe next e.g. expected improvement, probability of improvement and upper confidence bound [10], there are few well known strategies for selecting batches of points. To the best of our knowledge, every batch selection

strategy proposed in the literature involves a *greedy* algorithm, which chooses individual points until the batch is filled. Greedy choice making can be severely detrimental, for example, a greedy approach to the travelling salesman problem could potentially lead to the uniquely worst global solution [11]. In this work, our key contribution is to provide what we believe is the first non-greedy algorithm to choose a batch of points to probe next in the task of parallel global optimization.

Our approach is to choose a set of points which in expectation, maximally reduces our uncertainty about the location of the maximizer of the objective function. The algorithm we develop, *parallel predictive entropy search*, extends the methods of [12, 13] to multiple point batch selection. In Section 2, we formalize the problem and discuss previous approaches before developing parallel predictive entropy search in Section 3. Finally, we demonstrate the benefit of our non-greedy strategy on synthetic as well as real-world objective functions in Section 4.

## 2    Problem Statement and Background

Our aim is to maximize an objective function $f : \mathcal{X} \to \mathbb{R}$, which is unknown but can be (noisily) evaluated pointwise at multiple locations in parallel. In this work, we assume $\mathcal{X}$ is a compact subset of $\mathbb{R}^D$. At each decision, we must select a set of $Q$ points $\mathcal{S}_t = \{\boldsymbol{x}_{t,1}, ..., \boldsymbol{x}_{t,Q}\} \subset \mathcal{X}$, where the objective function would next be evaluated in parallel. Each evaluation leads to a scalar observation $y_{t,q} = f(\boldsymbol{x}_{t,q}) + \epsilon_{t,q}$, where we assume $\epsilon_{t,q} \sim \mathcal{N}(0, \sigma^2)$ i.i.d. We wish to minimize a future *regret*, $r_T = [f(\boldsymbol{x}^*) - f(\tilde{\boldsymbol{x}}_T)]$, where $\boldsymbol{x}^* \in \mathrm{argmax}_{\boldsymbol{x} \in \mathcal{X}} f(\boldsymbol{x})$ is an optimal decision (assumed to exist) and $\tilde{\boldsymbol{x}}_T$ is our guess of where the maximizer of $f$ is after evaluating $T$ batches of input points. It is highly intractable to make decisions $T$ steps ahead in the setting described, therefore it is common to consider the regret of the very next decision. In this work, we shall assume $f$ is a draw from a Gaussian process with constant mean $\lambda \in \mathbb{R}$ and differentiable kernel function $k : \mathcal{X}^2 \to \mathbb{R}$.

Most Bayesian optimization research focuses on choosing a single point to query at each decision i.e. $Q = 1$. A popular strategy in this setting is to choose the point with highest expected improvement over the current best evaluation, i.e. the maximizer of $a_{\mathrm{EI}}(\boldsymbol{x}|\mathcal{D}) = \mathbb{E}\big[\max(f(\boldsymbol{x}) - f(\boldsymbol{x}_{\mathrm{best}}), 0)|\mathcal{D}\big] = \sigma(\boldsymbol{x})\big[\phi\big(\tau(\boldsymbol{x})\big) + \tau(\boldsymbol{x})\Phi\big(\tau(\boldsymbol{x})\big)\big]$, where $\mathcal{D}$ is the set of observations, $\boldsymbol{x}_{\mathrm{best}}$ is the best evaluation point so far, $\sigma(\boldsymbol{x}) = \sqrt{\mathrm{Var}[f(\boldsymbol{x})|\mathcal{D}]}$, $\mu(\boldsymbol{x}) = \mathbb{E}[f(\boldsymbol{x})|\mathcal{D}]$, $\tau(\boldsymbol{x}) = (\mu(\boldsymbol{x}) - f(\boldsymbol{x}_{\mathrm{best}}))/\sigma(\boldsymbol{x})$ and $\phi(.)$ and $\Phi(.)$ are the standard Gaussian p.d.f. and c.d.f.

Aside from being an intuitive approach, a key advantage of using the *expected improvement* strategy is in the fact that it is computable analytically and is infinitely differentiable, making the problem of finding $\mathrm{argmax}_{\boldsymbol{x} \in \mathcal{X}} a_{\mathrm{EI}}(\boldsymbol{x}|\mathcal{D})$ amenable to a plethora of gradient based optimization methods. Unfortunately, the corresponding strategy for selecting $Q > 1$ points to evaluate in parallel does not lead to an analytic expression. [14] considered an approach which sequentially used the EI criterion to greedily choose a batch of points to query next, which [3] formalized and utilized by defining

$$a_{\mathrm{EI-MCMC}}\big(\boldsymbol{x}|\mathcal{D}, \{\boldsymbol{x}_{q'}\}_{q'=1}^q\big) = \int_{\mathcal{X}^q} a_{\mathrm{EI}}\big(\boldsymbol{x}|\mathcal{D} \cup \{\boldsymbol{x}_{q'}, y_{q'}\}_{q'=1}^q\big) p\big(\{y_{q'}\}_{q'=1}^q|\mathcal{D}, \{\boldsymbol{x}_{q'}\}_{q'=1}^q\big) dy_1..dy_q,$$

the expected gain in evaluating $\boldsymbol{x}$ after evaluating $\{\boldsymbol{x}_{q'}, y_{q'}\}_{q'=1}^q$, which can be approximated using Monte Carlo samples, hence the name EI-MCMC. Choosing a batch of points $\mathcal{S}_t$ using the EI-MCMC policy is *doubly greedy*: (i) the EI criterion is greedy as it inherently aims to minimize one-step regret, $r_t$, and (ii) the EI-MCMC approach starts with an empty set and populates it sequentially (and hence greedily), deciding the best single point to include until $|\mathcal{S}_t| = Q$.

A similar but different approach called *simulated matching* (SM) was introduced by [15]. Let $\pi$ be a baseline policy which chooses a single point to evaluate next (e.g. EI). SM aims to select a batch $\mathcal{S}_t$ of size $Q$, which includes a point 'close to' the best point which $\pi$ would have chosen when applied sequentially $Q$ times, with high probability. Formally, SM aims to maximize

$$a_{\mathrm{SM}}(\mathcal{S}_t|\mathcal{D}) = -\mathbb{E}_{S_\pi^Q}\Big[\mathbb{E}_f\Big[\min_{\boldsymbol{x} \in \mathcal{S}_t}(\boldsymbol{x} - \mathrm{argmax}_{\boldsymbol{x}' \in S_\pi^Q} f(\boldsymbol{x}'))^2 \Big| \mathcal{D}, S_\pi^Q\Big]\Big],$$

where $S_\pi^Q$ is the set of $Q$ points which policy $\pi$ would query if employed sequentially. A greedy $k$-medoids based algorithm is proposed to approximately maximize the objective, which the authors justify by the submodularity of the objective function.

The *upper confidence bound* (UCB) strategy [16] is another method used by practitioners to decide where to evaluate an objective function next. The UCB approach is to maximize $a_{\mathrm{UCB}}(\boldsymbol{x}|\mathcal{D}) = \mu(\boldsymbol{x}) + \alpha_t^{1/2}\sigma(\boldsymbol{x})$, where $\alpha_t$ is a domain-specific time-varying positive parameter which trades off

exploration and exploitation. In order to extend this approach to the parallel setting, [17] noted that the predictive variance of a Gaussian process depends only on where observations are made, and not the observations themselves. Therefore, they suggested the GP-BUCB method, which greedily populates the set $\mathcal{S}_t$ by maximizing a UCB type equation $Q$ times sequentially, updating $\sigma$ at each step, whilst maintaining the same $\mu$ for each batch. Finally, a variant of the GP-UCB was proposed by [18]. The first point of the set $\mathcal{S}_t$ is chosen by optimizing the UCB objective. Thereafter, a 'relevant region' $\mathcal{R}_t \subset \mathcal{X}$ which contains the maximizer of $f$ with high probability is defined. Points are greedily chosen from this region to maximize the information gain about $f$, measured by expected reduction in entropy, until $|\mathcal{S}_t| = Q$. This method was named Gaussian process upper confidence bound with pure exploration (GP-UCB-PE).

Each approach discussed resorts to a greedy batch selection process. To the best of our knowledge, no batch Bayesian optimization method to date has avoided a greedy algorithm. We avoid a greedy batch selection approach with PPES, which we develop in the next section.

## 3 Parallel Predictive Entropy Search

Our approach is to maximize information [19] about the location of the global maximizer $\boldsymbol{x}^*$, which we measure in terms of the negative differential entropy of $p(\boldsymbol{x}^*|\mathcal{D})$. Analogous to [13], PPES aims to choose the set of $Q$ points, $\mathcal{S}_t = \{\boldsymbol{x}_q\}_{q=1}^Q$, which maximizes

$$a_{\text{PPES}}(\mathcal{S}_t|\mathcal{D}) = \text{H}\big[p(\boldsymbol{x}^*|\mathcal{D})\big] - \mathbb{E}_{p\big(\{y_q\}_{q=1}^Q\big|\mathcal{D},\mathcal{S}_t\big)}\Big[\text{H}\big[p(\boldsymbol{x}^*|\mathcal{D} \cup \{\boldsymbol{x}_q, y_q\}_{q=1}^Q)\big]\Big], \qquad (1)$$

where $\text{H}[p(\boldsymbol{x})] = -\int p(\boldsymbol{x}) \log p(\boldsymbol{x}) d\boldsymbol{x}$ is the differential entropy of its argument and the expectation above is taken with respect to the posterior joint predictive distribution of $\{y_q\}_{q=1}^Q$ given the previous evaluations, $\mathcal{D}$, and the set $\mathcal{S}_t$. Evaluating (1) exactly is typically infeasible. The prohibitive aspects are that $p\big(\boldsymbol{x}^*|\mathcal{D} \cup \{\boldsymbol{x}_q, y_q\}_{q=1}^Q\big)$ would have to be evaluated for many different combinations of $\{\boldsymbol{x}_q, y_q\}_{q=1}^Q$, and the entropy computations are not analytically tractable in themselves. Significant approximations need to be made to (1) before it becomes practically useful [12]. A convenient equivalent formulation of the quantity in (1) can be written as the mutual information between $\boldsymbol{x}^*$ and $\{y_q\}_{q=1}^Q$ given $\mathcal{D}$ [20]. By symmetry of the mutual information, we can rewrite $a_{\text{PPES}}$ as

$$a_{\text{PPES}}(\mathcal{S}_t|\mathcal{D}) = \text{H}\big[p(\{y_q\}_{q=1}^Q|\mathcal{D},\mathcal{S}_t)\big] - \mathbb{E}_{p(\boldsymbol{x}^*|\mathcal{D})}\Big[\text{H}\big[p(\{y_q\}_{q=1}^Q|\mathcal{D},\mathcal{S}_t,\boldsymbol{x}^*)\big]\Big], \qquad (2)$$

where $p\big(\{y_q\}_{q=1}^Q|\mathcal{D},\mathcal{S}_t,\boldsymbol{x}^*\big)$ is the joint posterior predictive distibution for $\{y_q\}_{q=1}^Q$ given the observed data, $\mathcal{D}$ and the location of the global maximizer of $f$. The key advantage of the formulation in (2), is that the objective is based on entropies of predictive distributions of the observations, which are much more easily approximated than the entropies of distributions on $\boldsymbol{x}^*$.

In fact, the first term of (2) can be computed analytically. Suppose $p\big(\{f_q\}_{q=1}^Q|\mathcal{D},\mathcal{S}_t\big)$ is multi-variate Gaussian with covariance $\mathbf{K}$, then $\text{H}\big[p\big(\{y_q\}_{q=1}^Q|\mathcal{D},\mathcal{S}_t\big)\big] = 0.5 \log[\det(2\pi e(\mathbf{K} + \sigma^2\mathbf{I}))]$. We develop an approach to approximate the expectation of the predictive entropy in (2), using an expectation propagation based method which we discuss in the following section.

### 3.1 Approximating the Predictive Entropy

Assuming a sample of $\boldsymbol{x}^*$, we discuss our approach to approximating $\text{H}\big[p\big(\{y_q\}_{q=1}^Q|\mathcal{D},\mathcal{S}_t,\boldsymbol{x}^*\big)\big]$ in (2) for a set of query points $\mathcal{S}_t$. Note that we can write

$$p\big(\{y_q\}_{q=1}^Q|\mathcal{D},\mathcal{S}_t,\boldsymbol{x}^*\big) = \int p\big(\{f_q\}_{q=1}^Q|\mathcal{D},\mathcal{S}_t,\boldsymbol{x}^*\big) \prod_{q=1}^Q p(y_q|f_q) \ df_1...df_Q, \qquad (3)$$

where $p\big(\{f_q\}_{q=1}^Q|\mathcal{D},\mathcal{S}_t,\boldsymbol{x}^*\big)$ is the posterior distribution of the objective function at the locations $\boldsymbol{x}_q \in \mathcal{S}_t$, given previous evaluations $\mathcal{D}$, and that $\boldsymbol{x}^*$ is the global maximizer of $f$. Recall that $p(y_q|f_q)$ is Gaussian for each $q$. Our approach will be to derive a Gaussian approximation to $p\big(\{f_q\}_{q=1}^Q|\mathcal{D},\mathcal{S}_t,\boldsymbol{x}^*\big)$, which would lead to an analytic approximation to the integral in (3).

The posterior predictive distribution of the Gaussian process, $p\big(\{f_q\}_{q=1}^Q|\mathcal{D},\mathcal{S}_t\big)$, is multivariate Gaussian distributed. However, by further conditioning on the location $\boldsymbol{x}^*$, the global maximizer of $f$, we impose the condition that $f(\boldsymbol{x}) \leq f(\boldsymbol{x}^*)$ for any $\boldsymbol{x} \in \mathcal{X}$. Imposing this constraint for

all $\boldsymbol{x} \in \mathcal{X}$ is extremely difficult and makes the computation of $p\big(\{f_q\}_{q=1}^{Q}|\mathcal{D}, \mathcal{S}_t, \boldsymbol{x}^*\big)$ highly intractable. We instead impose the following two conditions (i) $f(\boldsymbol{x}) \leq f(\boldsymbol{x}^\star)$ for each $\boldsymbol{x} \in \mathcal{S}_t$, and (ii) $f(\boldsymbol{x}^\star) \geq y_{\max} + \epsilon$, where $y_{\max}$ is the largest observed noisy objective function value and $\epsilon \sim \mathcal{N}(0, \sigma^2)$. Constraint (i) is equivalent to imposing that $f(\boldsymbol{x}^\star)$ is larger than objective function values at current query locations, whilst condition (ii) makes $f(\boldsymbol{x}^\star)$ larger than previous objective function evaluations, accounting for noise. Denoting the two conditions $\mathcal{C}$, and the variables $\boldsymbol{f} = [f_1, ..., f_Q]^\top$ and $\boldsymbol{f}_+ = [\boldsymbol{f}; f^\star]$, where $f^\star = f(\boldsymbol{x}^*)$, we incorporate the conditions as follows

$$p\big(\boldsymbol{f}|\mathcal{D}, \mathcal{S}_t, \boldsymbol{x}^*\big) \approx \int p\big(\boldsymbol{f}_+|\mathcal{D}, \mathcal{S}_t, \boldsymbol{x}^*\big) \Phi\Big(\frac{f^\star - y_{\max}}{\sigma}\Big) \prod_{q=1}^{Q} \mathbb{I}(f^\star \geq f_q) \, df^\star, \qquad (4)$$

where $\mathbb{I}(.)$ is an indicator function. The integral in (4) can be approximated using expectation propagation [21]. The Gaussian process predictive $p\big(\boldsymbol{f}_+|\mathcal{D}, \mathcal{S}_t, \boldsymbol{x}^*\big)$ is $\mathcal{N}(\boldsymbol{f}_+; \mathbf{m}_+, \mathbf{K}_+)$. We approximate the integrand of (4) with $w(\boldsymbol{f}_+) = \mathcal{N}(\boldsymbol{f}_+; \mathbf{m}_+, \mathbf{K}_+) \prod_{q=1}^{Q+1} \tilde{Z}_q \mathcal{N}(\boldsymbol{c}_q^\top \boldsymbol{f}_+; \tilde{\mu}_q, \tilde{\tau}_q)$, where each $\tilde{Z}_q$ and $\tilde{\tau}_q$ are positive, $\tilde{\mu}_q \in \mathbb{R}$ and for $q \leq Q$, $\boldsymbol{c}_q$ is a vector of length $Q+1$ with $q^{\text{th}}$ entry $-1$, $Q + 1^{\text{st}}$ entry 1, and remaining entries 0, whilst $\boldsymbol{c}_{Q+1} = [0, ..., 0, 1]^\top$. The approximation $w(\boldsymbol{f}_+)$ approximates the Gaussian CDF, $\Phi(.)$, and each indicator function, $\mathbb{I}(.)$, with a univariate, scaled Gaussian PDF. The *site parameters*, $\{\tilde{Z}_q, \tilde{\mu}_q, \tilde{\tau}_q\}_{q=1}^{Q+1}$, are learned using a fast EP algorithm, for which details are given in the supplementary material, where we show that $w(\boldsymbol{f}_+) = Z\mathcal{N}(\boldsymbol{f}_+; \boldsymbol{\mu}_+, \boldsymbol{\Sigma}_+)$, where

$$\boldsymbol{\mu}_+ = \boldsymbol{\Sigma}_+ \bigg(\mathbf{K}_+^{-1}\mathbf{m}_+ + \sum_{q=1}^{Q+1} \frac{\tilde{\mu}_q}{\tilde{\tau}_q} \boldsymbol{c}_q \boldsymbol{c}_q^\top\bigg)^{-1}, \qquad \boldsymbol{\Sigma}_+ = \bigg(\mathbf{K}_+^{-1} + \sum_{q=1}^{Q+1} \frac{1}{\tilde{\tau}_q} \boldsymbol{c}_q \boldsymbol{c}_q^\top\bigg)^{-1}, \qquad (5)$$

and hence $p\big(\boldsymbol{f}_+|\mathcal{D}, \mathcal{S}_t, \mathcal{C}\big) \approx \mathcal{N}(\boldsymbol{f}_+; \boldsymbol{\mu}_+, \boldsymbol{\Sigma}_+)$. Since multivariate Gaussians are consistent under marginalization, a convenient corollary is that $p\big(\boldsymbol{f}|\mathcal{D}, \mathcal{S}_t, \boldsymbol{x}^*\big) \approx \mathcal{N}(\boldsymbol{f}; \boldsymbol{\mu}, \boldsymbol{\Sigma})$, where $\boldsymbol{\mu}$ is the vector containing the first $Q$ elements of $\boldsymbol{\mu}_+$, and $\boldsymbol{\Sigma}$ is the matrix containing the first $Q$ rows and columns of $\boldsymbol{\Sigma}_+$. Since sums of independent Gaussians are also Gaussian distributed, we see that $p\big(\{y_q\}_{q=1}^{Q}|\mathcal{D}, \mathcal{S}_t, \boldsymbol{x}^*\big) \approx \mathcal{N}([y_1, ..., y_Q]^\top; \boldsymbol{\mu}, \boldsymbol{\Sigma} + \sigma^2\mathbf{I})$. The final convenient attribute of our Gaussian approximation, is that the differential entropy of a multivariate Gaussian can be computed analytically, such that $\mathrm{H}\big[p\big(\{y_q\}_{q=1}^{Q}|\mathcal{D}, \mathcal{S}_t, \boldsymbol{x}^*\big)\big] \approx 0.5 \log[\det(2\pi e(\boldsymbol{\Sigma} + \sigma^2\mathbf{I}))]$.

## 3.2 Sampling from the Posterior over the Global Maximizer

So far, we have considered how to approximate $\mathrm{H}\big[p\big(\{y_q\}_{q=1}^{Q}|\mathcal{D}, \mathcal{S}_t, \boldsymbol{x}^*\big)\big]$, given the global maximizer, $\boldsymbol{x}^*$. We in fact would like the expected value of this quantity over the posterior distribution of the global maximizer, $p(\boldsymbol{x}^*|\mathcal{D})$. Literally, $p(\boldsymbol{x}^*|\mathcal{D}) \equiv p(f(\boldsymbol{x}^*) = \max_{\boldsymbol{x} \in \mathcal{X}} f(\boldsymbol{x})|\mathcal{D})$, the posterior probability that $\boldsymbol{x}^*$ is the global maximizer of $f$. Computing the distribution $p(\boldsymbol{x}^*|\mathcal{D})$ is intractable, but it is possible to approximately sample from it and compute a Monte Carlo based approximation of the desired expectation. We consider two approaches to sampling from the posterior of the global maximizer: (i) a maximum a posteriori (MAP) method, and (ii) a random feaure approach.

**MAP sample from $p(\boldsymbol{x}^*|\mathcal{D})$.** The MAP of $p(\boldsymbol{x}^*|\mathcal{D})$ is its posterior mode, given by $\boldsymbol{x}^*_{\text{MAP}} = \text{argmax}_{\boldsymbol{x}^* \in \mathcal{X}} p(\boldsymbol{x}^*|\mathcal{D})$. We may approximate the expected value of the predictive entropy by replacing the posterior distribution of $\boldsymbol{x}^*$ with a single point estimate at $\boldsymbol{x}^*_{\text{MAP}}$. There are two key advantages to using the MAP estimate in this way. Firstly, it is simple to compute $\boldsymbol{x}^*_{\text{MAP}}$, as it is the global maximizer of the posterior mean of $f$ given the observations $\mathcal{D}$. Secondly, choosing to use $\boldsymbol{x}^*_{\text{MAP}}$ assists the EP algorithm developed in Section 3.1 to converge as desired. This is because the condition $f(\boldsymbol{x}^*) \geq f(\boldsymbol{x})$ for $\boldsymbol{x} \in \mathcal{X}$ is easy to enforce when $\boldsymbol{x}^* = \boldsymbol{x}^*_{\text{MAP}}$, the global maximizer of the poserior mean of $f$. When $\boldsymbol{x}^*$ is sampled such that the posterior mean at $\boldsymbol{x}^*$ is significantly suboptimal, the EP approximation may be poor. Whilst using the MAP estimate approximation is convenient, it is after all a point estimate and fails to characterize the full posterior distribution. We therefore consider a method to draw samples from $p(\boldsymbol{x}^*|\mathcal{D})$ using random features.

**Random Feature Samples from $p(\boldsymbol{x}^*|\mathcal{D})$.** A naive approach to sampling from $p(\boldsymbol{x}^*|\mathcal{D})$ would be to sample $g \sim p(f|\mathcal{D})$, and choosing $\text{argmax}_{\boldsymbol{x} \in \mathcal{X}} g$. Unfortunately, this would require sampling $g$ over an uncountably infinite space, which is infeasible. A slightly less naive method would be to sequentially construct $g$, whilst optimizing it, instead of evaluating it everywhere in $\mathcal{X}$. However, this approach would have cost $\mathcal{O}(m^3)$ where $m$ is the number of function evaluations of $g$ necessary to find its optimum. We propose as in [13], to sample and optimize an analytic approximation to $g$.

By Bochner's theorem [22], a stationary kernel function, $k$, has a Fourier dual $s(\boldsymbol{w})$, which is equal to the spectral density of $k$. Setting $p(\boldsymbol{w}) = s(\boldsymbol{w})/\alpha$, a normalized density, we can write

$$k(\boldsymbol{x}, \boldsymbol{x}') = \alpha \mathbb{E}_{p(\boldsymbol{w})}[e^{-i\boldsymbol{w}^\top(\boldsymbol{x}-\boldsymbol{x}')}] = 2\alpha \mathbb{E}_{p(\boldsymbol{w},b)}[\cos(\boldsymbol{w}^\top \boldsymbol{x} + b)\cos(\boldsymbol{w}^\top \boldsymbol{x}' + b)], \qquad (6)$$

where $b \sim U[0, 2\pi]$. Let $\boldsymbol{\phi}(\boldsymbol{x}) = \sqrt{2\alpha/m}\cos(\mathbf{W}\boldsymbol{x} + \boldsymbol{b})$ denote an $m$-dimensional feature mapping where $\mathbf{W}$ and $\boldsymbol{b}$ consist of $m$ stacked samples from $p(\boldsymbol{w}, b)$, then the kernel $k$ can be approximated by the inner product of these features, $k(\boldsymbol{x}, \boldsymbol{x}') \approx \boldsymbol{\phi}(\boldsymbol{x})^\top \boldsymbol{\phi}(\boldsymbol{x}')$ [23]. The linear model $g(\boldsymbol{x}) = \boldsymbol{\phi}(\boldsymbol{x})^\top \boldsymbol{\theta} + \lambda$ where $\boldsymbol{\theta}|\mathcal{D} \sim \mathcal{N}(\mathbf{A}^{-1}\boldsymbol{\phi}^\top(\boldsymbol{y} - \lambda\mathbf{1}), \sigma^2\mathbf{A}^{-1})$ is an approximate sample from $p(f|\mathcal{D})$, where $\boldsymbol{y}$ is a vector of objective function evaluations, $\mathbf{A} = \boldsymbol{\phi}^\top\boldsymbol{\phi} + \sigma^2\mathbf{I}$ and $\boldsymbol{\phi}^\top = [\boldsymbol{\phi}(\boldsymbol{x}_1)...\boldsymbol{\phi}(\boldsymbol{x}_n)]$. In fact, $\lim_{m\to\infty} g$ is a true sample from $p(f|\mathcal{D})$ [24].

The generative process above suggests the following approach to approximately sampling from $p(\boldsymbol{x}^*|\mathcal{D})$: (i) sample random features $\boldsymbol{\phi}^{(i)}$ and corresponding posterior weights $\boldsymbol{\theta}^{(i)}$ using the process above, (ii) construct $g^{(i)}(\boldsymbol{x}) = \boldsymbol{\phi}^{(i)}(\boldsymbol{x})^\top \boldsymbol{\theta}^{(i)} + \lambda$, and (iii) finally compute $\boldsymbol{x}^{\star(i)} = \operatorname{argmax}_{\boldsymbol{x}\in\mathcal{X}} g^{(i)}(\boldsymbol{x})$ using gradient based methods.

### 3.3 Computing and Optimizing the PPES Approximation

Let $\boldsymbol{\psi}$ denote the set of kernel parameters and the observation noise variance, $\sigma^2$. Our posterior belief about $\boldsymbol{\psi}$ is summarized by the posterior distribution $p(\boldsymbol{\psi}|\mathcal{D}) \propto p(\boldsymbol{\psi})p(\mathcal{D}|\boldsymbol{\psi})$, where $p(\boldsymbol{\psi})$ is our prior belief about $\boldsymbol{\psi}$ and $p(\mathcal{D}|\boldsymbol{\psi})$ is the GP marginal likelihood given the parameters $\boldsymbol{\psi}$. For a fully Bayesian treatment of $\boldsymbol{\psi}$, we must marginalize $a_{\text{PPES}}$ with respect to $p(\boldsymbol{\psi}|\mathcal{D})$. The expectation with respect to the posterior distribution of $\boldsymbol{\psi}$ is approximated with Monte Carlo samples. A similar approach is taken in [3, 13]. Combining the EP based method to approximate the predictive entropy with either of the two methods discussed in the previous section to approximately sample from $p(\boldsymbol{x}^*|\mathcal{D})$, we can construct $\hat{a}_{\text{PPES}}$ an approximation to (2), defined by

$$\hat{a}_{\text{PPES}}(\mathcal{S}_t|\mathcal{D}) = \frac{1}{2M}\sum_{i=1}^{M}\Big[\log[\det(\mathbf{K}^{(i)} + \sigma^{2(i)}\mathbf{I})] - \log[\det(\boldsymbol{\Sigma}^{(i)} + \sigma^{2(i)}\mathbf{I})]\Big], \qquad (7)$$

where $\mathbf{K}^{(i)}$ is constructed using $\boldsymbol{\psi}^{(i)}$ the $i^{\text{th}}$ sample of $M$ from $p(\boldsymbol{\psi}|\mathcal{D})$, $\boldsymbol{\Sigma}^{(i)}$ is constructed as in Section 3.1, assuming the global maximizer is $\boldsymbol{x}^{*(i)} \sim p(\boldsymbol{x}^*|\mathcal{D}, \boldsymbol{\psi}^{(i)})$. The PPES approximation is simple and amenable to gradient based optimization. Our goal is to choose $\mathcal{S}_t = \{\boldsymbol{x}_1, ..., \boldsymbol{x}_Q\}$ which maximizes $\hat{a}_{\text{PPES}}$ in (7). Since our kernel function is differentiable, we may consider taking the derivative of $\hat{a}_{\text{PPES}}$ with respect to $x_{q,d}$, the $d^{\text{th}}$ component of $\boldsymbol{x}_q$,

$$\frac{\partial \hat{a}_{\text{PPES}}}{\partial x_{q,d}} = \frac{1}{2M}\sum_{i=1}^{M}\left[\operatorname{trace}\Big[(\mathbf{K}^{(i)} + \sigma^{2(i)}\mathbf{I})^{-1}\frac{\partial \mathbf{K}^{(i)}}{\partial x_{q,d}}\Big] - \operatorname{trace}\Big[(\boldsymbol{\Sigma}^{(i)} + \sigma^{2(i)}\mathbf{I})^{-1}\frac{\partial \boldsymbol{\Sigma}^{(i)}}{\partial x_{q,d}}\Big]\right]. \quad (8)$$

Computing $\frac{\partial \mathbf{K}^{(i)}}{\partial x_{q,d}}$ is simple directly from the definition of the chosen kernel function. $\boldsymbol{\Sigma}^{(i)}$ is a function of $\mathbf{K}^{(i)}$, $\{\boldsymbol{c}_q\}_{q=1}^{Q+1}$ and $\{\tilde{\sigma}_q^{(i)}\}_{q=1}^{Q+1}$, and we know how to compute $\frac{\partial \mathbf{K}^{(i)}}{\partial x_{q,d}}$, and that each $\boldsymbol{c}_q$ is a constant vector. Hence our only concern is how the EP site parameters, $\{\tilde{\sigma}_q^{(i)}\}_{q=1}^{Q+1}$, vary with $x_{q,d}$. Rather remarkably, we may invoke a result from Section 2.1 of [25], which says that converged site parameters, $\{\tilde{Z}_q, \tilde{\mu}_q, \tilde{\sigma}_q\}_{q=1}^{Q+1}$, have 0 derivative with respect to parameters of $p(\boldsymbol{f}_+|\mathcal{D}, \mathcal{S}_t, \boldsymbol{x}^*)$. There is a key distinction between *explicit* dependencies (where $\boldsymbol{\Sigma}$ actually depends on $\mathbf{K}$) and *implicit* dependencies where a site parameter, $\tilde{\sigma}_q$, might depend implicitly on $\mathbf{K}$. A similar approach is taken in [26], and discussed in [7]. We therefore compute

$$\frac{\partial \boldsymbol{\Sigma}_+^{(i)}}{\partial x_{q,d}} = \boldsymbol{\Sigma}_+^{(i)}\mathbf{K}_+^{(i)-1}\frac{\partial \mathbf{K}_+^{(i)}}{\partial x_{q,d}}\mathbf{K}_+^{(i)-1}\boldsymbol{\Sigma}_+^{(i)}. \qquad (9)$$

On first inspection, it may seem computationally too expensive to compute derivatives with respect to each $q$ and $d$. However, note that we may compute and store the matrices $\mathbf{K}_+^{(i)-1}\boldsymbol{\Sigma}_+^{(i)}$, $(\mathbf{K}^{(i)} + \sigma^{2(i)}\mathbf{I})^{-1}$ and $(\boldsymbol{\Sigma}^{(i)} + \sigma^{2(i)}\mathbf{I})^{-1}$ once, and that $\frac{\partial \mathbf{K}_+^{(i)}}{\partial x_{q,d}}$ is symmetric with exactly one non-zero row and non-zero column, which can be exploited for fast matrix multiplication and trace computations.

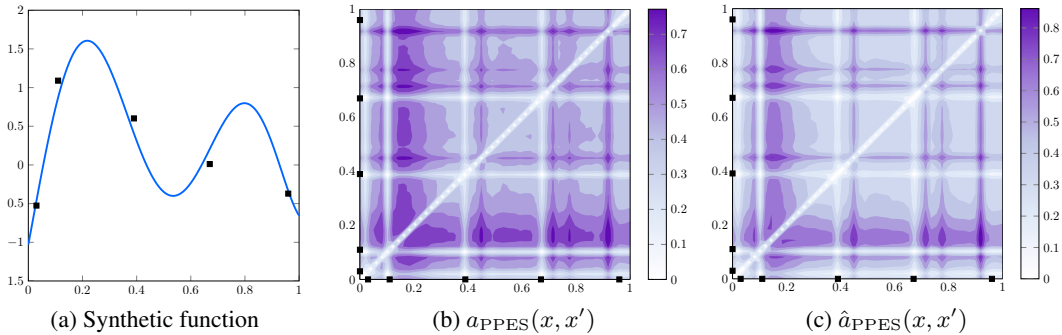

(a) Synthetic function  (b) $a_{\text{PPES}}(x, x')$  (c) $\hat{a}_{\text{PPES}}(x, x')$

Figure 1: Assessing the quality of our approximations to the parallel predictive entropy search strategy. (a) Synthetic objective function (blue line) defined on $[0, 1]$, with noisy observations (black squares). (b) Ground truth $a_{\text{PPES}}$ defined on $[0, 1]^2$, obtained by rejection sampling. (c) Our approximation $\hat{a}_{\text{PPES}}$ using expectation propagation. Dark regions correspond to pairs $(x, x')$ with high utility, whilst faint regions correspond to pairs $(x, x')$ with low utility.

## 4  Empirical Study

In this section, we study the performance of PPES in comparison to aforementioned methods. We model $f$ as a Gaussian process with constant mean $\lambda$ and covariance kernel $k$. Observations of the objective function are considered to be independently drawn from $\mathcal{N}(f(\boldsymbol{x}), \sigma^2)$. In our experiments, we choose to use a squared-exponential kernel of the form $k(\boldsymbol{x}, \boldsymbol{x}') = \gamma^2 \exp\left[-0.5\sum_d (x_d - x'_d)^2/l_d^2\right]$. Therefore the set of model hyperparameters is $\{\lambda, \gamma, l_1, ..., l_D, \sigma\}$, a broad Gaussian hyperprior is placed on $\lambda$ and uninformative Gamma priors are used for the other hyperparameters.

It is worth investigating how well $\hat{a}_{\text{PPES}}$ (7) is able to approximate $a_{\text{PPES}}$ (2). In order to test the approximation in a manner amenable to visualization, we generate a sample $f$ from a Gaussian process prior on $\mathcal{X} = [0, 1]$, with $\gamma^2 = 1$, $\sigma^2 = 10^{-4}$ and $l^2 = 0.025$, and consider batches of size $Q = 2$. We set $M = 200$. A rejection sampling based approach is used to compute the ground truth $a_{\text{PPES}}$, defined on $\mathcal{X}^Q = [0, 1]^2$. We first discretize $[0, 1]^2$, and sample $p(\boldsymbol{x}^*|\mathcal{D})$ in (2) by evaluating samples from $p(f|\mathcal{D})$ on the discrete points and choosing the input with highest function value. Given $\boldsymbol{x}^*$, we compute $\text{H}\left[p(y_1, y_2|\mathcal{D}, \boldsymbol{x}_1, \boldsymbol{x}_2, \boldsymbol{x}^*)\right]$ using rejection sampling. Samples from $p(f|\mathcal{D})$ are evaluted on discrete points in $[0, 1]^2$ and rejected if the highest function value occurs not at $\boldsymbol{x}^*$. We add independent Gaussian noise with variance $\sigma^2$ to the non rejected samples from the previous step and approximate $\text{H}\left[p(y_1, y_2|\mathcal{D}, \boldsymbol{x}_1, \boldsymbol{x}_2, \boldsymbol{x}^*)\right]$ using kernel density estimation [27].

Figure 1 includes illustrations of (a) the objective function to be maximized, $f$, with 5 noisy observations, (b) the $a_{\text{PPES}}$ ground truth obtained using the rejection sampling method and finally (c) $\hat{a}_{\text{PPES}}$ using the EP method we develop in the previous section. The black squares on the axes of Figures 1(b) and 1(c) represent the locations in $\mathcal{X} = [0, 1]$ where $f$ has been noisily sampled, and the darker the shade, the larger the function value. The lightly shaded horizontal and vertical lines in these figures along the points The figures representing $a_{\text{PPES}}$ and $\hat{a}_{\text{PPES}}$ appear to be symmetric, as is expected, since the set $\mathcal{S}_t = \{x, x'\}$ is not an ordered set, since all points in the set are probed in parallel i.e. $\mathcal{S}_t = \{x, x'\} = \{x', x\}$. The surface of $\hat{a}_{\text{PPES}}$ is similar to that of $a_{\text{PPES}}$. In paticular, the $\hat{a}_{\text{PPES}}$ approximation often appeared to be an *annealed* version of the ground truth $a_{\text{PPES}}$, in the sense that peaks were more pronounced, and non-peak areas were flatter. Since we are interested in $\text{argmax}_{\{x,x'\} \in \mathcal{X}^2} a_{\text{PPES}}(\{x, x'\})$, our key concern is that the peaks of $\hat{a}_{\text{PPES}}$ occur at the same input locations as $a_{\text{PPES}}$. This appears to be the case in our experiment, suggesting that the $\text{argmax}\,\hat{a}_{\text{PPES}}$ is a good approximation for $\text{argmax}\,a_{\text{PPES}}$.

We now test the performance of PPES in the task of finding the optimum of various objective functions. For each experiment, we compare PPES ($M = 200$) to EI-MCMC (with 100 MCMC samples), simulated matching with a UCB baseline policy, GP-BUCB and GP-UCB-PE. We use the random features method to sample from $p(\boldsymbol{x}^*|\mathcal{D})$, rejecting samples which lead to failed EP runs. An experiment of an objective function, $f$, consists of sampling 5 input points uniformly at random and running each algorithm starting with these samples and their corresponding (noisy) function values. We measure performance after $t$ batch evaluations using *immediate regret*, $r_t = |f(\tilde{\boldsymbol{x}}_t) - f(\boldsymbol{x}^*)|$, where $\boldsymbol{x}^*$ is the known optimizer of $f$ and $\tilde{\boldsymbol{x}}_t$ is the recommendation of an algorithm after $t$ batch evaluations. We perform 100 experiments for each objective function, and report the median of the

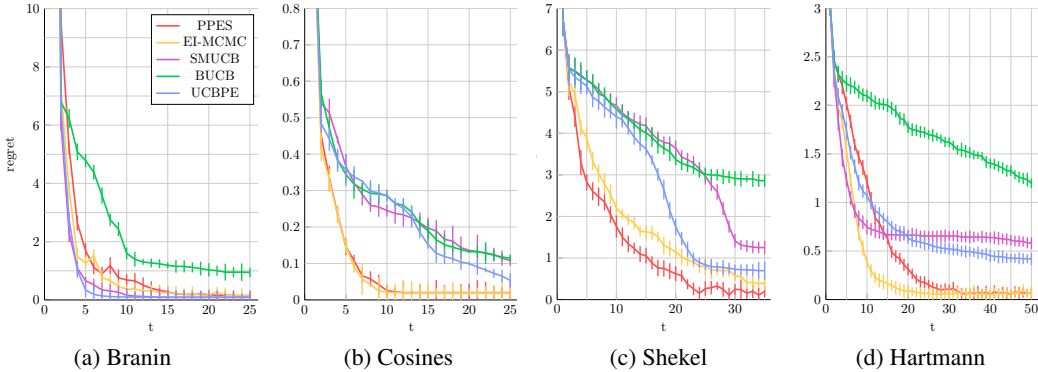

| (a) Branin | (b) Cosines | (c) Shekel | (d) Hartmann |

Figure 2: Median of the immediate regret of the PPES and 4 other algorithms over 100 experiments on benchmark synthetic objective functions, using batches of size $Q = 3$.

immediate regret obtained for each algorithm. The confidence bands represent one standard deviation obtained from bootstrapping. The empirical distribution of the immediate regret is heavy tailed, making the median more representative of where most data points lie than the mean.

Our first set of experiments is on a set of synthetic benchmark objective functions including Branin-Hoo [28], a mixture of cosines [29], a Shekel function with 10 modes [30] (each defined on $[0, 1]^2$) and the Hartmann-6 function [28] (defined on $[0, 1]^6$). We choose batches of size $Q = 3$ at each decision time. The plots in Figure 2 illustrate the median immediate regrets found for each algorithm. The results suggest that the PPES algorithm performs close to best if not the best for each problem considered. EI-MCMC does significantly better on the Hartmann function, which is a relatively smooth function with very few modes, where greedy search appears beneficial. Entropy-based strategies are more exploratory in higher dimensions. Nevertheless, PPES does significantly better than GP-UCB-PE on 3 of the 4 problems, suggesting that our non-greedy batch selection procedure enhances performance versus a greedy entropy based policy.

We now consider maximization of real world objective functions. The first, `boston`, returns the negative of the prediction error of a neural network trained on a random train/text split of the Boston Housing dataset [31]. The weight-decay parameter and number of training iterations for the neural network are the parameters to be optimized over. The next function, `hydrogen`, returns the amount of hydrogen produced by particular bacteria as a function of pH and nitrogen levels of a growth medium [32]. Thirdly we consider a function, `rocket`, which runs a simulation of a rocket [33] being launched from the Earth's surface and returns the time taken for the rocket to land on the Earth's surface. The variables to be optimized over are the launch height from the surface, the mass of fuel to use and the angle of launch with respect to the Earth's surface. If the rocket does not return, the function returns 0. Finally we consider a function, `robot`, which returns the walking speed of a bipedal robot [34]. The function's input parameters, which live in $[0, 1]^8$, are the robot's controller. We add Gaussian noise with $\sigma = 0.1$ to the noiseless function. Note that all of the functions we consider are not available analytically. `boston` trains a neural network and returns test error, whilst `rocket` and `robot` run physical simulations involving differential equations before returning a desired quantity. Since the hydrogen dataset is available only for discrete points, we define `hydrogen` to return the predictive mean of a Gaussian process trained on the dataset.

Figure 3 show the median values of immediate regret by each method over 200 random initializations. We consider batches of size $Q = 2$ and $Q = 4$. We find that PPES consistently outperforms competing methods on the functions considered. The greediness and nonrequirement of MCMC sampling of the SM-UCB, GP-BUCB and GP-UCB-PE algorithms make them amenable to large batch experiments, for example, [17] consider optimization in $\mathbb{R}^{45}$ with batches of size 10. However, these three algorithms all perform poorly when selecting batches of smaller size. The performance on the `hydrogen` function illustrates an interesting phenemona; whilst the immediate regret of PPES is mediocre initially, it drops rapidly as more batches are evaluated.

This behaviour is likely due to the non-greediness of the approach we have taken. EI-MCMC makes good initial progress, but then fails to explore the input space as well as PPES is able to. Recall that after each batch evaluation, an algorithm is required to output $\tilde{x}_t$, its best estimate for the maximizer of the objective function. We observed that whilst competing algorithms tended to evaluate points which had high objective function values compared to PPES, yet when it came to recommending $\tilde{x}_t$,

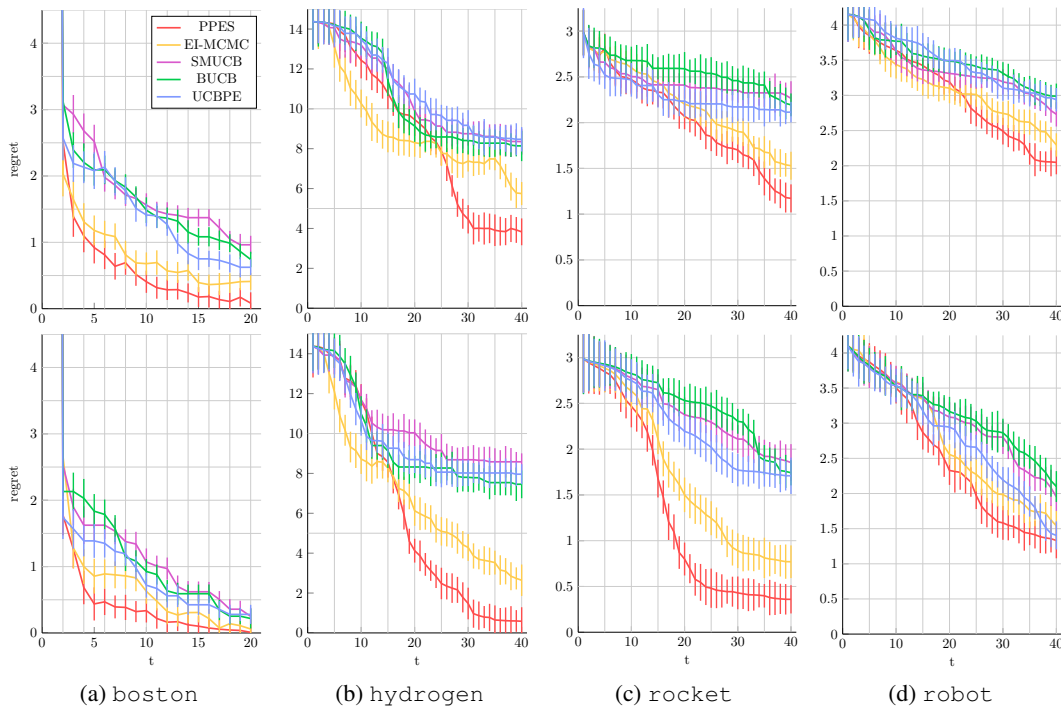

<div style="text-align:center">

(a) `boston`      (b) `hydrogen`      (c) `rocket`      (d) `robot`

</div>

Figure 3: Median of the immediate regret of the PPES and 4 other algorithms over 100 experiments on real world objective functions. Figures in the top row use batches of size $Q = 2$, whilst figues on the bottom row use batches of size $Q = 4$.

PPES tended to do a better job. Our belief is that this occured exactly because the PPES objective aims to maximize information gain rather than objective function value improvement.

The `rocket` function has a strong discontinuity making if difficult to maximize. If the fuel mass, launch height and/or angle are too high, the rocket would not return to the Earth's surface, resulting in a 0 function value. It can be argued that a stationary kernel Gaussian process is a poor model for this function, yet it is worth investigating the performance of a GP based models since a practitioner may not know whether or not their black-box function is smooth apriori. PPES seemed to handle this function best and had fewer samples which resulted in 0 function value than each of the competing methods and made fewer recommendations which led to a 0 function value. The relative increase in PPES performance from increasing batch size from $Q = 2$ to $Q = 4$ is small for the `robot` function compared to the other functions considered. We believe this is a consequence of using a slightly naive optimization procedure to save computation time. Our optimization procedure first computes $\hat{a}_{\text{PPES}}$ at 1000 points selected uniformly at random, and performs gradient ascent from the best point. Since $\hat{a}_{\text{PPES}}$ is defined on $\mathcal{X}^Q = [0, 1]^{32}$, this method may miss a global optimum. Other methods all select their batches greedily, and hence only need to optimize in $\mathcal{X} = [0, 1]^8$. However, this should easily be avoided by using a more exhaustive gradient based optimizer.

## 5 Conclusions

We have developed parallel predictive entropy search, an information theoretic approach to batch Bayesian optimization. Our method is greedy in the sense that it aims to maximize the one-step information gain about the location of $x^*$, but it is not greedy in how it selects a set of points to evaluate next. Previous methods are doubly greedy, in that they look one step ahead, and also select a batch of points greedily. Competing methods are prone to under exploring, which hurts their perfomance on multi-modal, noisy objective functions, as we demonstrate in our experiments.

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
