[Supplementary Material]

# Parallel Predictive Entropy Search for Batch Global Optimization of Expensive Objective Functions Supplementary Material

**Amar Shah**
Department of Engineering
Cambridge University
as793@cam.ac.uk

**Zoubin Ghahramani**
Department of Engineering
University of Cambridge
zoubin@eng.cam.ac.uk

In this document, we include details of the EP algorithm discussed in the main text.

Recall that $\boldsymbol{f} = [f_1, ..., f_Q]^\top$ and $\boldsymbol{f}_+ = [\boldsymbol{f}; f^\star]$, where $f^\star = f(\boldsymbol{x}^*)$ and $\boldsymbol{x}^*$ is the global maximizer of $f$. By being a Gaussian process predictive distribution, we know that $p(\boldsymbol{f}_+|\mathcal{D}, \mathcal{S}_t, \boldsymbol{x}^*)$ follows a multivariate Gaussian distribution of the form $\mathcal{N}(\boldsymbol{f}_+; \mathbf{m}_+, \mathbf{K}_+)$.

We impose two conditions, (i) that $f(\boldsymbol{x}^*)$ is larger than $f(\boldsymbol{x})$ for each $\boldsymbol{x}$ in the query set $\mathcal{S}_t$ and (ii) that $f(\boldsymbol{x}^*)$ is larger than previous observations, accounting for Gaussian noise. We denote the conditions $\mathcal{C}$. Our goal is to make a Gaussian approximation to

$$p(\boldsymbol{f}_+|\mathcal{D}, \mathcal{S}_t, \mathcal{C}) \propto p(\boldsymbol{f}_+|\mathcal{D}, \mathcal{S}_t, \boldsymbol{x}^*)\Phi\Big(\frac{f^\star - y_{\max}}{\sigma}\Big)\prod_{q=1}^{Q}\mathbb{I}(f^\star \geq f_q). \tag{1}$$

An elegant approach to making such a Gaussian approximation, is by using expectation propagation. We approximate the term involving $\Phi(.)$ and each $\mathbb{I}(.)$ with a univariate scaled Gaussian p.d.f. such that our unnormalized approximation to $p(\boldsymbol{f}_+|\mathcal{D}, \mathcal{S}_t, \mathcal{C})$ is

$$w(\boldsymbol{f}_+) = \mathcal{N}(\boldsymbol{f}_+; \mathbf{m}_+, \mathbf{K}_+)\prod_{q=1}^{Q+1}\tilde{Z}_q\mathcal{N}(\boldsymbol{c}_q^\top \boldsymbol{f}_+; \tilde{\mu}_q, \tilde{\tau}_q), \tag{2}$$

where each $\tilde{Z}_q$ and $\tilde{\tau}_q$ is positive, $\tilde{\mu}_q \in \mathbb{R}$ and for $q \leq Q$, $\boldsymbol{c}_q$ is a vector of length $Q + 1$ with $q^{\text{th}}$ entry $-1$, $Q + 1^{\text{st}}$ entry 1, and remaining entries 0, whilst $\boldsymbol{c}_{Q+1} = [0, ..., 0, 1]^\top$. We have approximated each indicator function and Gaussian c.d.f. with a scaled Gaussian p.d.f. The *site parameters*, $\{\tilde{Z}_q, \tilde{\mu}_q, \tilde{\tau}_q\}_{q=1}^{Q+1}$, are to be optimized such that the Kullback-Leibler divergence of $w(\boldsymbol{f}_+)/\int w(\boldsymbol{f}_+')d\boldsymbol{f}_+'$ from $p(\boldsymbol{f}_+|\mathcal{D}, \mathcal{S}_t, \mathcal{C})$ is minimized.

Since products of Gaussian p.d.f.s lead to Gaussian p.d.f.s, $w(\boldsymbol{f}_+) = Z\mathcal{N}(\boldsymbol{f}_+; \boldsymbol{\mu}_+, \boldsymbol{\Sigma}_+)$, where

$$\boldsymbol{\mu}_+ = \boldsymbol{\Sigma}_+\Big(\mathbf{K}_+^{-1}\mathbf{m}_+ + \sum_{q=1}^{Q+1}\frac{\tilde{\mu}_q}{\tilde{\tau}_q}\boldsymbol{c}_q\boldsymbol{c}_q^\top\Big)^{-1}, \tag{3}$$

$$\boldsymbol{\Sigma}_+ = \Big(\mathbf{K}_+^{-1} + \sum_{q=1}^{Q+1}\frac{1}{\tilde{\tau}_q}\boldsymbol{c}_q\boldsymbol{c}_q^\top\Big)^{-1}, \tag{4}$$

$$\begin{aligned}
\log Z = &-\frac{1}{2}\big(\mathbf{m}_+\mathbf{K}_+^{-1}\mathbf{m}_+ + \log|\mathbf{K}_+|\big) \\
&+ \sum_{q=1}^{Q}\Big(\log\tilde{Z}_q - \frac{1}{2}\Big(\frac{\mu_q^2}{\tau_q} + \log\sigma_q^2 + \log(2\pi)\Big)\Big) \\
&+ \frac{1}{2}\big(\boldsymbol{\mu}_+\boldsymbol{\Sigma}_+^{-1}\boldsymbol{\mu}_+ + \log|\boldsymbol{\Sigma}_+|\big).
\end{aligned} \tag{5}$$

We now describe the steps required to update the site parameters. We closely follow the derivations in [1]. We first compute the *cavity* distributions,

$$w^{\backslash q}(\boldsymbol{f}_+) = \frac{w(\boldsymbol{f}_+)}{\tilde{Z}_q \mathcal{N}(\boldsymbol{c}_q^\top \boldsymbol{f}_+; \tilde{\mu}_q, \tilde{\tau}_q)} \tag{6}$$

and compute their Gaussian parameters. Since we are dividing a Gaussian p.d.f. by another Gaussian p.d.f. we have simple parameter updates given by

$$\tau_{\backslash q} = \left( (\boldsymbol{c}_q^\top \boldsymbol{\Sigma}_+^{-1} \boldsymbol{c}_q)^{-1} - \tilde{\tau}_q^{-1} \right)^{-1} \tag{7}$$

$$\mu_{\backslash q} = \tau_{\backslash q} \left( \frac{\boldsymbol{c}_q^\top \boldsymbol{\mu}_+}{\boldsymbol{c}_q^\top \boldsymbol{\Sigma}_+ \boldsymbol{c}_q} - \frac{\tilde{\mu}_q}{\tilde{\tau}_q} \right). \tag{8}$$

The next step of EP is the *projection* step and requires moment matching $\tilde{Z}_q \mathcal{N}(\boldsymbol{c}_q^\top \boldsymbol{f}_+; \tilde{\mu}_q, \tilde{\tau}_q) w^{\backslash q}(\boldsymbol{f}_+)$ with $t_q(\boldsymbol{f}_+) w^{\backslash q}(\boldsymbol{f}_+)$, where $t_q(\boldsymbol{f}_+)$ is the true $q^{\text{th}}$ factor being approximated. We use derivatives of the logarithm of the zeroth moment [2] to compute the parameters

$$\hat{Z}_q = \int t_q(\boldsymbol{f}_+) w^{\backslash q}(\boldsymbol{f}_+) d\boldsymbol{f}_+$$
$$= \Phi(\beta_q), \tag{9}$$

$$\hat{\mu}_q = \mu_{\backslash q} + \tau_{\backslash q} \frac{\partial \log \hat{Z}_q}{\partial \mu_{\backslash q}}$$
$$= \mu_{\backslash q} + \sqrt{\tau_{\backslash q}} \frac{\phi(\beta_q)}{\Phi(\beta_q)}, \tag{10}$$

$$\hat{\tau}_q = \tau_{\backslash q} - \mu_{\backslash q}^2 \left( \left( \frac{\partial \log \hat{Z}_q}{\partial \mu_{\backslash q}} \right)^2 - 2 \frac{\partial \log \hat{Z}_q}{\partial \tau_{\backslash q}} \right)$$
$$= \tau_{\backslash q} - \tau_{\backslash q} \left( \frac{\phi(\beta_q)}{\Phi(\beta_q)} \right) \left( \frac{\phi(\beta_q)}{\Phi(\beta_q)} + \beta_q \right), \tag{11}$$

where $\beta_q = \frac{\mu_{\backslash q}}{\sqrt{\tau_{\backslash q}}}$ for $q \leq Q$ and $\beta_{Q+1} = \Phi\left( \frac{\mu_{\backslash q} - y_{\max}}{\sqrt{\sigma^2 + \tau_{\backslash q}}} \right)$. To complete the projection step, we update the site parameters to achieve the moments computed above by setting

$$\tilde{\tau}_q = \left( \hat{\tau}_q^{-1} - \tau_{\backslash q}^{-1} \right)^{-1}, \tag{12}$$

$$\tilde{\mu}_q = \tilde{\tau}_q \left( \hat{\tau}_q^{-1} \hat{\mu}_q - \tau_{\backslash q}^{-1} \mu_{\backslash q} \right)^{-1}. \tag{13}$$

$$\tilde{Z}_q = \hat{Z}_q \sqrt{2\pi} \sqrt{\tau_{\backslash q} + \tilde{\tau}_q} \exp\left[ \frac{1}{2} \frac{\left( \mu_{\backslash q} - \tilde{\mu}_q \right)^2}{\left( \tau_{\backslash q} + \tilde{\tau}_q \right)} \right]. \tag{14}$$

Finally we update the parameters $\boldsymbol{\mu}_+$ and $\boldsymbol{\Sigma}_+$ as in equations (3) and (4), and repeat the proces until convergence.