[Reviews · NeurIPS 2015]

Submitted by Assigned_Reviewer_1

Paper Summary: Motivated by the optimization of objective functions which are extremely expensive to evaluate, but which can be accessed in parallel, the authors construct an algorithm for selecting batches of several instances to evaluate, called Parallel Predictive Entropy Search, or PPES.

PPES is an extension of the Predictive Entropy Search (PES) algorithm of Hernandez-Lobato et al., 2014.

PPES is constructed such that the instances which compose a batch jointly (i.e., non-greedily) optimize an approximate expected conditional mutual information between the corresponding observations and the location of the global optimizer of the objective function. In order to accomplish this, the authors develop an approximation to the mutual information using EP, construct an approximate sampler from the posterior over the global maximizer's location, and address how the optimization over the approximate expected conditional mutual information (i.e., the selection of the instances for the batch) can be performed.

They then proceed to test their approximate expected conditional mutual information in a simple setting.

Finally, they test the whole algorithm on a variety of data sets against several competing batch algorithms.

Quality: The approach taken by the authors is rigorous, sensible, and produces useful results.

There are several approximations, but they are well-motivated.

The numerical experiments are quite complete, though they could do with a bit more information in the supplement as to precisely what code was being run.

The approximation which most interests me is the one taken in Equation 4; essentially, the authors approximate p(f|D,S_t,x*) by a PDF conditioned on a series of necessary (f(x) < f(x*)) or quasi-necessary (f(x*) >= ymax + eps) conditions; how does this affect the entropy of the resulting distribution?

Intuitively, I would expect that conditioning only on necessary conditions would produce an entropy which is at least as large as the original, but what about the quasi-necessary conditioning?

Should we expect the information gain to be consistently under-estimated?

Is it reasonable to expect that the quasi-necessary condition won't bias the location of the maximizer?

Some intuition from the authors might be helpful here and in other places.

Clarity: The material and algorithm are both quite complicated, but the presentation is well done and this explanation is fairly clear.

Section 3, the major technical section, is particularly well constructed.

In regard to the experiments, it would be best if the implementations of the competing algorithms were discussed a bit, perhaps in the supplementary material; some of the algorithms are not explicitly designed to give a most probable maximizer, and the code for others (GP-BUCB, at least, and perhaps the Azimi et al. simulation matching algorithms) are unfortunately publicly available only in a form which takes discrete input sets; without having this information in some form, it's a bit hard to assess how much value the experiments have.

PPES looks to perform quite well, clearly, but how strong are the baselines?

Of course, the authors of this work should not be responsible for fixing the code of others, but some information on this would be handy.

Originality: The primary basis for the paper is Hernandez-Lobato, Hoffman, and Ghahramani 2014.

Relative to that work, which introduced PES, the PPES algorithm is a parallelized version that allows the selection of batches of actions.

Practically, this is an important new capability.

So far as I can tell, the primary theoretical differences are the removal of a condition from the approximation of the predictive entropy (conditioned on x*) in section 3.1, which appear to make this computation similar, and some differences in computing the gradient of the approximate decision function in section 3.3.

The EP implementation may also be substantially different, but this is somewhat hard to tell from the text.

These innovations do not appear to be enormous, or fundamental changes of approach.

Undoubtedly, there was plenty of work which went into extending PES to PPES, but the authors have not done a good job of highlighting precisely what the theoretical novelties of the present work are, or why they were necessary or interesting.

Significance: This work appears to be a good fit for the setting of controlling computationally intensive simulations, and will likely make a good impact there.

In particular, in this setting the method's high computational cost would be still very small as compared to the large simulations it would be controlling, the parallelism it presumes is available, and the experiments are very expensive.

I would argue that this combination of strong writing, good mathematical meat, and the availability of a well-suited problem domain make this a significant paper for practice.

That said, I can imagine that this setting would also often have batch sizes which are larger than the 2-4 attempted in this paper, and as the authors point out, their algorithm scales poorly for these cases.

How does this cost compare against the computational expense of a single observation in a reasonable setting, e.g., the robot problem addressed here? At what point is it reasonable to run a simpler algorithm to allow greater parallelism?

Again, some practical guidance would be helpful.
Summary: The authors develop PPES, an extension of the PES algorithm from strictly sequential decision-making to parallel decision-making for expensive objective functions.

This extension requires some apparently small technical changes, but the fundamental machinery of PES is kept intact.

This involves a series of well-motivated and reasonable approximations to the objective function, an expected information gain with respect to the location of the global optimizer. PPES seeks to jointly optimize this over the set of instances to be queried in the batch.

The work is of high quality, and clearly presented.

Though the technical improvements from PES to PPES appear small, the practical benefit of extension to the parallel setting seems substantial, though the batch sizes remain fairly small.

The authors conclude with a strong series of experiments, which show PPES performing well against a variety of competitors.

Submitted by Assigned_Reviewer_2

This paper presents an extension of predictive entropy search (PES) to perform non-greedy batch selection, which differs from the existing batch selection strategies that greedily chooses individual points. Its performance is evaluated using numerous synthetic and real-world datasets.

There are three serious concerns regarding the novelty and significance of the authors' proposed work:

(1) The presentation of the criterion and approximations of parallel PES (PPES) appears to be largely similar to that of PES. It is not clear whether PPES is a straightforward extension from PES or whether some important new technique/trick has been introduced to make PPES possible. During the course of reading the paper, I was expecting some novel treatment to scale up to a large batch Q. To my dismay, it was mentioned in the last 2 lines of this paper that such a crucial issue is not being addressed in this paper.

(2) Though PPES differs from the existing batch selection strategies that are greedy in nature, its choice of being non-greedy entails some limitations as compared to them, which are neither discussed nor addressed: (a) PPES does not scale well in the batch size Q and (b) no performance guarantee is given with respect to its approximations and assumptions. With regards to (b), the authors have only empirically demonstrated the closeness of its approximation in a simple toy example with Q=2.

(3) Given a fixed sampling budget, how does the batch size Q affect the trade-off between the optimization performance and time efficiency? This is not demonstrated empirically. How does the value of information play a role in this trade-off?

The following statement "Entropy-based strategies are more exploratory in higher dimensions." is not justified.

Minor issues:

Page 6: The line "The lightly shaded horizontal and vertical lines in these figures along the points" does not read well.

There are some grammatical errors.

It is not clear whether each tested experimental setting requires batch selection.
Summary: There are three serious concerns with this paper: (1) Similarity to the original predictive entropy search in terms of criterion and approximations, (2) poor scalability to large batch size and lack of performance guarantee as compared to existing batch selection strategies, and (3) no empirical study of the tradeoff between optimization performance and time efficiency with varying batch size given a fixed sampling budget.

Submitted by Assigned_Reviewer_3

The works presents a method for parallel Bayesian optimisation based on a non-greedy extension of the Predictive Entropy Search method (PES), Hernandez-Lobato et al. 2014. The problem is interesting and the methodology novel and well developed. Overall, I think that this is a good paper that should be accepted. I have, however, a few comments for the authors:

- Regarding the current literature in Batch Bayesian Optimisation methods, the authors claim that they propose the first non greedy BO algorithm. I think that this is not completely true since the Expected improvement has been extended for multiple (and joint) batch selection before. See http://hal-emse.ccsd.cnrs.fr/hal-00260579/document

- The proposed method is a non-greedy approach, which is a very elegant solution to the problem of batch selection. I think, however, that the authors have not discussed enough which are the advantages of non-greedy vs. greedy approaches in this context, which somehow justifies their solution to the problem. For example, one could say in defence of the greedy approaches that

the resulting surrogate optimisation problem is typically easier (lower dimensional) than the greedy ones, which obviously affects the scalability. Also, in some cases, it is interesting to truncate the batch collection when a saturation effect is produced in terms of the collected information. This is easy in non greedy approaches, but not doable (in principle) in greedy ones. The authors may argument that the main advantage is shown in the experimental section, where the results show that the PPES is the global best method. This is true. However, this shows the advantage of the method itself (loss function + batch selection) rather than the utility of using a greedy strategy. How much of that gain is due to use a non-greedy search method and how much due to the fact that it is based on the 'right' (PES) loss function? To quantify this, it may be interesting for the authors to compare their current method with a non-greedy version of it (whatever this is) and also with the baseline sequential PPES.

Summary: The paper presents a natural extansion to the work of Hernandez-Lobato et al 2014 on Predictive Entropy Search to the selection of batches of points. The paper is interesting, well written an technically correct. The experimental section is convicing. In my opinion this work should be accepted.

Author Feedback
Author rebuttal: We would like to thank the reviewers for their insights and suggestions.

@Reviewer 1:
It is interesting to study the impact on the imposed conditions on the actual entropy, but as you mention, it is most crucial to see the behavior of the argmax of the approximation, which is what we focused on with limited space.

@Reviewer2:
The PPES approximation is indeed similar to the PES approximation, the key complication which we dealt with was to compute the entropy of (f_1,...,f_Q)|(f^* > f_1,...,f_Q, x^*). When Q=1 this can be done exactly, which is the approach in PES. In PPES, we approximate these conditions and the condition that (f^*>y_max + e) jointly using EP. PES makes one approximation after another in series.

The problem with large Q becomes a problem of high dimensional optimization faced throughout machine learning. Although we showed the advantages of our method rather conservatively for small batches, we see no reason why if the application were to warrant it, it would be infeasible to optimize the acquisition function for large Q (analogous to how many ML problems such as training neural nets involve optimization in thousands or millions of dimensions). Note also that one pass of block gradient ascent in the QD-dimensional space is computationally just as cheap as Q steps in D-dimensions each. Moreover in many applications people care about small batch sizes, e.g. in wet-labs when you have 3 machines to run experiments on or in Formula 1 where you have 2 wind tunnels to test parts in parallel.

You are correct that we did not prove any performance guarantees of the method. We believe that no theoretical bounds have been shown for ES or PES either, so it would be an interesting research direction to start with proving bounds for these methods.

We chose Q=2 for our visualization example, since it's harder to visualize in higher dimensions.

The PPES method is comparable in speed to the EI-MCMC method, since the EP approximations are very fast to compute and the limiting factor is the sampling of hyperparameters. We can try to include examples of the performance against wall clock time in the supplementary material - the charts generally look the same.

@Reviewer 3:
Thank you for the pointer to the Multi-points Criterion paper, we were unaware of it. We like your suggestion of comparing PPES to greedy-batch PES to directly assess the benefit of being non-greedy, we should be able to implement this test for the camera ready version.

@Reviewer 6:
The MCMC method is not a big issue actually. All we are required to do is sample hyperparameters given observations at each time step, which is actually not too bad when we warm-start the slice sampler with previously used samples. Since EP converges very fast relative to sampling, the runtime of PPES is similar to that of EI-MCMC.